# Tephrosin Suppresses the Chemoresistance of Paclitaxel-Resistant Ovarian Cancer via Inhibition of FGFR1 Signaling Pathway

**DOI:** 10.3390/biomedicines11123155

**Published:** 2023-11-27

**Authors:** Hee Su Kim, Sowon Bae, Ye Jin Lim, Kyeong A So, Tae Jin Kim, Seunghee Bae, Jae Ho Lee

**Affiliations:** 1Department of Cosmetics Engineering, Konkuk University, 120 Neungdong-ro, Gwangjin-gu, Seoul 05029, Republic of Korea; kjhumy2@naver.com (H.S.K.); bsw4136@gmail.com (S.B.); yejin513513@naver.com (Y.J.L.); sbae@konkuk.ac.kr (S.B.); 2Department of Obstetrics and Gynecology, Konkuk University, 120 Neungdong-ro, Gwangjin-gu, Seoul 05030, Republic of Korea; joyfulplace@hanmail.net (K.A.S.); kimonc111@naver.com (T.J.K.)

**Keywords:** ovarian cancer, paclitaxel resistance, tephrosin, FGFR1/FRS2 signaling pathway

## Abstract

Ovarian cancer is the leading cause of death among gynecologic cancers. Paclitaxel is used as a standard first-line therapeutic agent for ovarian cancer. However, chemotherapeutic resistance and high recurrence rates are major obstacles to treating ovarian cancer. We have found that tephrosin, a natural rotenoid isoflavonoid, can resensitize paclitaxel-resistant ovarian cancer cells to paclitaxel. Cell viability, immunoblotting, and a flow cytometric analysis showed that a combination treatment made up of paclitaxel and tephrosin induced apoptotic death. Tephrosin inhibited the phosphorylation of AKT, STAT3, ERK, and p38 MAPK, all of which simultaneously play important roles in survival signaling pathways. Notably, tephrosin downregulated the phosphorylation of FGFR1 and its specific adapter protein FRS2, but it had no effect on the phosphorylation of the EGFR. Immunoblotting and a fluo-3 acetoxymethyl assay showed that tephrosin did not affect the expression or function of P-glycoprotein. Additionally, treatment with N-acetylcysteine did not restore cell cytotoxicity caused by a treatment combination made up of paclitaxel and tephrosin, showing that tephrosin did not affect the reactive oxygen species scavenging pathway. Interestingly, tephrosin reduced the expression of the anti-apoptotic factor XIAP. This study demonstrates that tephrosin is a potent antitumor agent that can be used in the treatment of paclitaxel-resistant ovarian cancer via the inhibition of the FGFR1 signaling pathway.

## 1. Introduction

Epithelial ovarian cancer is one of the most lethal gynecological cancers [1,2]. It has been reported that the initial diagnosis rate of ovarian cancer is low, and less than half of patients survive more than five years after diagnosis [3]. First-line therapies are based on cytoreductive surgery and are combined with platinum- and taxane-based chemotherapeutics [4,5,6]. Paclitaxel, one of the taxane families, binds to the β-subunit of tubulin in the absence of GTP, a typical factor that is essential for microtubule polymerization [7]. Binding stabilizes the associated microtubules and inhibits tubulin depolymerization [8,9]. Then, cell cycle arrest and apoptosis are induced [10]. The chemoresistance of tumor cells is a major cause of chemotherapy failure for most human cancers [11,12]. In particular, it has been reported that 80% of ovarian cancer patients experience a relapse of paclitaxel resistance [5,13,14], which demands the development of strategies to suppress chemotherapy resistance and enhance patient survival with improved treatment effectiveness.

Accumulating evidence has implied that various mechanisms have been identified that contribute to chemoresistance [12,15]. Although pharmacological factors due to an inadequate drug concentration and reduced drug stability at the tumor site may contribute to reduced therapeutic efficiency, cellular factors play an important role in chemoresistance [12,16]. It has been reported that hyperactivation related to the survival signaling pathway among cellular factors affects chemoresistance [17], including AKT [18,19,20], STAT3 [21], ERK [22,23], and p38 MAPK [24,25]. In particular, AKT has been identified to mediate the survival signals that preserve various cancer cells in the cell death pathway [15,26,27] and regulate sensitivity to paclitaxel [28,29,30]. ERK, STAT3, and p38 MAPK have also been found to regulate paclitaxel-induced chemosensitivity in many carcinomas, including lung cancer [31], gastric cancer [32], breast cancer [33,34], and ovarian cancer [35,36]. In addition to the cellular factors related to the prosurvival pathway, endogenous reactive oxygen species (ROS) contribute to the resistance to chemotherapy by activating the MAPK and AKT signaling pathways [37,38,39]. In addition, previous studies have identified that the regulation of the expression of pro- and anti-apoptotic factors is related to chemoresistance in cancer [40,41,42]. It has been reported that acquired drug resistance is also caused by increased drug efflux and decreased drug uptake [43]. P-gp (P-glycoprotein) plays a key role in drug efflux [9], and previous studies have shown that the regulation of P-gp expression mediates chemoresistance in various cancers [44,45,46].

The fibroblast growth factor receptor (FGFR) is a kind of receptor tyrosine kinase (RTK) encoded by four genes, namely FGFR1, FGFR2, FGFR3, and FGFR4 [47]. The binding of ligands to the FGFR family induces receptor dimerization and the phosphorylation of the tyrosine kinase domain [48]. As the docking protein FGFR substrate (FRS2) is activated, it binds to growth factor receptor-bound 2 (GRB2) or the tyrosine phosphatase SHP2 protein. The RAS/MAPK [49] and PI3K/AKT [50] signaling pathways are activated. In the case of the STAT pathway, it is activated by the phosphorylation of the FGFR [51]. As a physiological function of the FGFR signaling pathway, it is involved in the overall development process of major cells, such as proliferation, differentiation, and survival [52]. The FGFR signaling pathway also induces the proliferation, survival, migration, and invasion of cancer cells [53]. Dysregulation of the FGFR signaling pathway has been observed in various cancers, and amplifications of the FGFR gene account for 66% of all aberrations among FGFR alterations [48]. Helsten T et al. reported that FGFR aberrations occurred frequently in gynecological cancers, such as breast carcinoma, endometrial adenocarcinoma, and ovarian carcinoma, among all types of cancer [54]. Previous studies have reported that FRS2, a downstream factor of the FGFR signaling pathway, is repeatedly amplified in high-grade serous ovarian cancer (HGSOC), which accounts for 70–80% of all ovarian carcinomas [55]. FRS2 has been identified as one of 50 genes essential for survival in ovarian cancer cell lines [55]. Previous studies have shown that the overexpression of FRS2 promotes tumorigenesis in ovarian cancer [56], breast cancer [57], lung adenocarcinoma [58], and prostate cancer [59]. Regarding paclitaxel resistance, several studies have revealed the effect of resensitization by targeting the FGFR signaling pathway [60,61,62,63]. Despite research showing that the FGFR signaling pathway is important in cancer, the role of the FGFR signaling pathway in paclitaxel-resistant ovarian cancer is still not entirely clear.

Rotenone is a natural hydrophobic component mainly isolated from the root and bark of *Derris* species. As a highly toxic compound, rotenone has been used as a herbicide and an insecticide [64]. It has also been reported that rotenone exhibits antitumor activity in lung cancer [65], colon cancer [66], and breast cancer [67]. Tephrosin, one of the rotenoid families, has also been reported to have anticancer effects in some cancers, including pancreatic cancer [68] and non-small-cell lung cancer [69]. The anticancer effects of tephrosin are gradually being determined, but currently the effect of tephrosin on ovarian cancer is not clear. Furthermore, whether tephrosin affects paclitaxel resistance is not known. Here, we reported for the first time that tephrosin restores chemosensitivity in paclitaxel-resistant ovarian cancer cells via inhibiting the FGFR signaling pathway.

## 2. Materials and Methods

### 2.1. Cell Lines and Culture

SKOV3, SKOV3-TR, and HeyA8-MDR ovarian carcinoma cell lines were provided by Professor A.K. Sood (University of Texas MD Anderson Cancer Center, Houston, TX, USA). All cell lines were cultured in Roswell Park Memorial Institute’s (RPMI) 1640 medium (Biowest, Nuaillé, France), supplemented with 10% fetal bovine serum (FBS, Corning, NY, USA) and 1% penicillin/streptomycin (Thermo Fisher Scientific, Waltham, MA, USA). A humidified 37 °C incubator with 5% CO₂ was used to culture these cell lines. To maintain the paclitaxel resistance for SKOV3-TR and HeyA8-MDR cells, 50 nM of paclitaxel was added once every 2 days for sub-culture. Tephrosin and BGJ398 were purchased from MedChemExpress (Monmouth Junction, NJ, USA). N-acetylcysteine (NAC) was obtained from Sigma-Aldrich (St. Louis, MO, USA).

### 2.2. Cell Viability Using the Water-Soluble Tetrazolium 1 (WST-1) Assay and Crystal Violet Assay

Cells (5.0 × 10^3^) were seeded in 96-well plates for 24 h and then treated with each drug for 48 h. The WST-1 assay was performed to determine cell viability using EZ-Cytox (DoGen, Seoul, Republic of Korea). The medium contained in each well was suctioned and washed once with Dulbecco’s Phosphate-Buffered Salines (DPBS, WelGENE, Seoul, Republic of Korea). The EZ-Cytox solution was then diluted to 1/10 in RPMI 1640 medium and added to each well. After incubation for 30 min, cell viability analysis was performed by measuring the absorbance value at 450 nm using the Synergy™ HTX Multi-Mode Microplate Reader (Bioteck, Winooski, VT, USA). The crystal violet assay was examined to visually confirm the results of the WST-1 assay on cell viability of each drug. Cells (5.0 × 10^4^) were seeded in 24-well plates for 24 h and then treated with each drug for 48 h. Then, cells were added to 500 µL of 0.2% crystal violet solution (Biopure, Seoul, Republic of Korea) in each well and stained for 30 min.

### 2.3. Cell Cytotoxicity Analysis Using the Lactate Dehydrogenase (LDH) Assay

The LDH assay was used to identify cell cytotoxicity using the EZ-LDH kit (DoGen, Seoul, Republic of Korea). After 48 h of treatment with tephrosin (1, 2, 5, 10, 20, and 40 µM) in SKOV3-TR cells, the culture media were collected separately, and the floating cells were precipitated using centrifugation (600× *g*, 5 min). In total, 10 µL of the supernatant was added in triplicate to 100 µL of the LDH reaction mixture (WST substrate mix and LDH assay buffer) in new 96-well plates. The mixtures were reacted for 30 min at room temperature in the dark. Cell cytotoxicity was analyzed by measuring absorbance at 450 nm using the Synergy™ HTX Multi-Mode Microplate Reader (Bioteck, Winooski, VT, USA).

### 2.4. Reverse Transcription Polymerase Chain Reaction (RT-PCR) and Reverse Transcription Quantitative Polymerase Chain Reaction (RT-qPCR) Tests

Cells (2.0 × 10^5^) were seeded in 60 mm culture plates for 24 h. After 48 h of treatment with paclitaxel (200 nM), tephrosin (10 µM), and a combination treatment composed of paclitaxel and tephrosin, the total genomic RNA was extracted using the Ribo-EX reagent (GeneAll Biotechnology Co., Ltd., Seoul, Republic of Korea). Complementary DNA was synthesized from the RNA using M-MLV Reverse Transcriptase (Invitrogen, MA, USA). First-strand cDNA was included in the PCR amplification mixture. This mixture included a 2.5 mM dNTP mixture, 10X reaction buffer with MgCl_2_, Taq DNA polymerase (Bioneer, Daejeon, Republic of Korea), dimethyl sulfoxide (DMSO, Sigma-Aldrich, Darmstadt, Germany), and each specific primer mixture. PCR products were subjected to gel electrophoresis in 1% agarose (BioShop Canada Inc., Burlington, ON, Canada) containing StaySafe Nucleic Acid Gel Stain (Real Biotech Corporation, Banqiao City, Taiwan) to confirm the mRNA expression of the related genes. EvaGreen qPCR Mix Plus (ROX) master mix (Solis BioDyne, Tartu, Estonia) was contained in RT-qPCR, and the relative quantifications of target gene mRNA were analyzed using the StepOnePlus™ Real-Time PCR System (Applied Biosystems, Waltham, MA, USA). The primer sequences for RT-PCR and RT-qPCR tests are presented in Appendix A.

### 2.5. Immunoblotting

Cells (6.0 × 10^5^) were seeded in 100 mm culture plates without treatment with paclitaxel (50 nM) for 24 h. After 48 h of treatment with each drug, total cells were lysed in RIPA buffer (25 mM Tris-Cl, 150 mM NaCl, 1% NP40, 1% sodium deoxycholate, 0.025% SDS, 5 mM ethylenediaminetetraacetic acid) supplemented with PhosSTOP (Roche, Basel, Switzerland). The amount of total protein was quantified using a BCA Protein Assay Kit (Thermo Fisher Scientific, MA, USA). Proteins were separated using 8% sodium dodecyl sulfate polyacrylamide gel electrophoresis (SDS-PAGE). The proteins in the gel were transferred to 0.2 µm nitrocellulose membranes (Cytiva, Amersham, UK). After blotting, the membranes were blocked in 2% skim milk (Biopure, Cambridge, MA, USA) for 90 min and incubated with primary antibodies overnight at 4 °C.

Antibodies against Actin (sc-47778), FRS2 (sc-17841), XIAP (sc-55551), and caspase 9 (sc-17784) were purchased from Santa Cruz Biotechnology (Dallas, Texas, USA). Antibodies against cleaved PARP (#9541), AKT (#9272), phosphorylated AKT at S473 (#9271), STAT3 (#9139), phosphorylated STAT3 at Y705 (#9145), phosphorylated STAT3 at S727 (#92994), ERK (#9102), phosphorylated ERK at T202/T204 (#9101), p38 (#9212), phosphorylated p38 at T180/T182 (#9211), FGFR1 (#9740), phosphorylated FGFR1 at Y653/Y654 (#52928), phosphorylated FRS2 at Y196 (#3864), EGFR (#2232), phosphorylated EGFR at Y1045 (#2237), BCL-XL (#2764), MCL-1 (#5453), BAX (#5023), caspase 3 (#9665), cleaved caspase 3 (#9664), Survivin (#2803), and MDR1/ABCB1 (#12693S) were obtained from Cell Signaling Technology (Beverly, MA, USA). The antibody against BCL-2 (A19693) was purchased from ABclonal (Woburn, MA, USA), and the antibody against FGFR2 (PA5-14651) was obtained from Invitrogen (Waltham, MA, USA).

The membranes were incubated with secondary antibodies for 2 h. Anti-mouse IgG and anti-rabbit IgG were purchased from Cell Signaling Technology (Beverly, MA, USA). The protein expression levels were confirmed using the Clarity Western ECL Substrate (Biorad, CA, USA). The protein band intensities were quantified using ImageJ software from the National Institutes of Health (ImageJ 1.52a, Bethesda, MD, USA).

### 2.6. Fluorescence-Activated Cell Sorting (FACS) Analysis

A FACS analysis was examined to determine the effect of treatment with tephrosin and paclitaxel on cell cycle progression. SKOV3-TR cells (6.0 × 10^5^) were seeded in 100 mm plates for 24 h and then treated with two drugs. After 48 h of treatment, cells were washed twice with DPBS, pelleted, and fixed with 80% ethanol for 30 min. After being washed twice with cold DPBS, cells were incubated with 200 mg/mL RNase A (Qiagen, Hilden, German) for 30 min at 37 °C. Then, the cells were stained with 100 mg/mL propidium iodide (Sigma-Aldrich, Darmstadt, Germany) for 30 min at room temperature. The samples were immediately analyzed using FACS (BD Bioscience, Mountain View, CA, USA).

### 2.7. Fluo-3 Acetoxymethyl (AM) Assay

SKOV3 cells and SKOV3-TR cells (6.0 × 10^5^) were incubated for 24 h. Cells were treated with DMSO (mock), paclitaxel (200 nM), tephrosin (10 µM), and a combination treatment composed of paclitaxel and tephrosin for 48 h. Then, 4 µM fluo-3 acetoxylmethyl (AM) solution (Sigma-Aldrich, Darmstadt, Germany) was added and incubated for an additional 1 h at 37 °C.

### 2.8. Statistical Analysis

The experimental data were presented by expressing the mean ± standard deviation (SD) obtained from three distinct experiments. Significance in statistical variations was assigned to cases where the *p*-value ≤ 0.05. Furthermore, statistical differences were analyzed using a one-way analysis of variance (ANOVA), followed by Dunnett’s post hoc test.

## 3. Results

### 3.1. Combination Treatment with Paclitaxel and Tephrosin Effectively Induced Cytotoxicity in SKOV3-TR Cells

To examine whether cell cytotoxicity is affected by treatment with tephrosin alone in paclitaxel-resistant ovarian cancer SKOV3-TR cells, the LDH assay was used. Cells were treated with tephrosin (0, 1, 2, 5, 10, 20, and 40 µM) for 48 h. As shown in Figure 1A, the LDH assay showed that 4.3% cytotoxicity was observed when cells were treated with 20 µM of tephrosin. On the other hand, cytotoxicity did not appear at treatment concentrations below 10 µM. Next, we examined the effect of tephrosin on paclitaxel resistance using the WST-1 assay in SKOV3-TR cells. Cells were treated with serially diluted concentrations of paclitaxel (0–500 nM) together with dose-dependent concentrations of tephrosin (0, 5, and 10 µM) for 48 h. As shown in Figure 1B, the combination treatment with paclitaxel and tephrosin effectively decreased the cell viability of SKOV3-TR cells in a dose-dependent manner (*p* ≤ 0.05). The crystal violet analysis also showed that tephrosin induced cytotoxicity in paclitaxel-treated SKOV3-TR cells (Figure 1C). To identify whether the mechanism of tephrosin in paclitaxel resistance is related to cellular apoptosis, immunoblotting was performed to detect the expression of cleaved PARP. As shown in Figure 1D, SKOV3-TR cells were treated with paclitaxel and tephrosin for 24, 32, 40, and 48 h. Immunoblotting showed that cleaved PARP was not induced by treatment with paclitaxel and tephrosin alone. On the other hand, a combination treatment composed of paclitaxel and tephrosin increased the expression of cleaved PARP at 40 and 48 h (Figure 1D). As shown in Figure 1E, SKOV3-TR cells were cotreated with paclitaxel (0, 1, 10, 100, and 200 nM) and tephrosin (0, 1, 5, and 10 µM) for 48 h. Cleaved PARP expressions were increased by a combination treatment composed of paclitaxel and tephrosin in a dose-dependent manner. SKOV3-TR cells were treated with paclitaxel (200 nM), tephrosin (10 µM), and the paclitaxel and tephrosin combination treatment for 48 h. As shown in Figure 1F, microscopic observations showed that the combination treatment with paclitaxel and tephrosin increased cell toxicity in SKOV3-TR cells. To determine the fraction of the sub-G1 phase in SKOV3-TR cells, flow cytometry was performed (Figure 1G). The sub-G1 fraction phase of the combination treatment composed of paclitaxel and tephrosin was significantly increased compared to the other treatments (*p* ≤ 0.05) (2.65% in control, 4.83% in paclitaxel-treated, 7.62% in tephrosin-treated, and 36.00% in paclitaxel and tephrosin combination-treated). These data indicated that tephrosin effectively restored paclitaxel sensitivity in SKOV3-TR cells.

### 3.2. Tephrosin Inhibited the Phosphorylation of AKT, STAT3, ERK, and p38 MAPK in SKOV3-TR Cells

Previous studies have reported that intracellular signaling factors such as PI3K/AKT [70,71], STAT3 [21], ERK [72], and p38 MAPK [24] are closely related to cell proliferation and survival. Additionally, these factors are known to mediate the chemoresistance of various cancers [28,29,30,31,32,33,34,35,36]. To determine whether tephrosin affects the PI3K/AKT, STAT3, ERK, and p38 MAPK signaling pathways in SKOV3-TR cells, immunoblotting was examined. As a result of Figure 2, the phosphorylation levels of AKT (S473), STAT3 (Y705), ERK (T202/Y204), and p38 MAPK (T180/T182) were effectively inhibited in a treatment composed of tephrosin alone and in a combination treatment made up of paclitaxel and tephrosin. These data suggested the possibility that tephrosin reverses chemoresistance by downregulating the prosurvival signaling pathway of paclitaxel-resistant ovarian cancer cells.

### 3.3. Tephrosin Downregulated the Phosphorylation of FGFR1/FRS2 Signaling Pathway in SKOV3-TR Cells

An extensive number of studies have shown that RTKs play an important role in a variety of oncogenic processes, including the regulation of proliferation, motility, and metastasis [73,74]. Recently, RTK signaling was studied as one of the major mechanisms that regulate chemoresistance in various cancer cells, including breast cancer [75] and ovarian cancer [76]. We sought to investigate whether the activity of the EGFR and FGFR, as representative types of RTKs, is regulated by tephrosin, and we examined immunoblotting in SKOV3-TR cells treated with a combination of tephrosin and paclitaxel. In Figure 3A, treatment with tephrosin did not affect the expression of the EGFR or its phosphorylation at Y1045 residue in SKOV3-TR cells. Interestingly, treatment with tephrosin effectively inhibited the phosphorylation of FGFR1 at Y653/654 residues, and cotreatment with tephrosin and paclitaxel effectively decreased the expression level of FGFR1 (Figure 3A). In addition, the phosphorylation of FRS2, an adaptor protein of FGFR1, was also inhibited by tephrosin treatment in SKOV3-TR cells. We also verified the inhibition of FGFR1 signaling by tephrosin in SKOV3-TR cells using a selective pan-FGFR inhibitor, BGJ398. As shown in lane 5 and 6 of Figure 3A, BGJ398 treatment downregulated the phosphorylation of FGFR1 and its adaptor protein FRS2 in the same way as tephrosin, indicating that tephrosin effectively inhibits FGFR1 signaling in SKOV3-TR cells. Next, SKOV3-TR cells were cotreated with paclitaxel (0, 1, 10, 100, and 200 nM) and tephrosin (0, 1, 5, and 10 µM) for 48 h. Its expression and the phosphorylation levels of FGFR1 and FRS2 were examined using immunoblotting. FGFR1 signaling was effectively downregulated by the combination treatment with paclitaxel and tephrosin in a dose-dependent manner (Figure 3B). As shown in Figure 3A,B, treatment with tephrosin downregulated FGFR1 phosphorylation and the cellular protein levels simultaneously. RT-PCR and RT-qPCR tests were performed to confirm whether the decrease in the FGFR1 expression induced by tephrosin was at the transcriptional level. Figure 3C,D showed that treatment with tephrosin did not affect the transcriptional expression of *FGFR1* and *FRS2*. The difference in the gene expression of *FGFR1* and *FRS2* was non-significant for all experimental groups (*p* > 0.05), indicating that tephrosin downregulated FGFR1 phosphorylation and also inhibited receptor stability in SKOV3-TR cells.

### 3.4. Tephrosin Suppressed Paclitaxel Resistance Independently of P-Glycoprotein (P-gp) Expression and Function in SKOV3-TR Cells

Accumulating amounts of evidence are showing that P-gp has substrates for several anticancer drugs, including paclitaxel and cisplatin, and its overexpression has been reported to be closely related to the chemoresistance of various cancers [44,45,46]. To investigate whether P-gp was overexpressed in SKOV3-TR cells compared to SKOV3 cells, immunoblotting was performed (Appendix A). The expression of P-gp was overexpressed in SKOV3-TR cells compared to SKOV3 cells. To determine whether tephrosin affects the expression of P-gp in SKOV3-TR cells, immunoblotting was analyzed. As shown in Figure 4A, the combination treatment composed of paclitaxel and tephrosin had no effect on the expression of P-gp in SKOV3-TR cells. It has been reported that the fluo-3/AM is a cell-permeable fluorescent molecule that acts as a substrate for P-gp and can be used to evaluate the function of P-gp [77]. In Figure 4B, SKOV3-TR cells did not show fluorescence after fluo-3/AM treatment compared to SKOV3 cells. Additionally, fluo-3/AM fluorescence was not observed in SKOV3-TR cells, even when tephrosin and paclitaxel were treated alone or together at 24 h and 48 h (Figure 4B,C). Collectively, these data suggested that the tephrosin-induced mechanism of restoring chemoresistance is not related to the expression and function of P-gp.

### 3.5. Tephrosin Inhibited the Expression of XIAP in SKOV3-TR Cells

Previous studies have reported that the regulation of endogenous ROS levels and the total antioxidant capacity of cells involves cancer proliferation, cellular apoptosis, and drug sensitivity [39,78,79]. To determine whether cellular apoptosis caused by the treatment combination composed of paclitaxel and tephrosin was mediated by intracellular ROS, N-acetylcysteine (NAC) was added in the treatment combination made up of paclitaxel and tephrosin in SKOV3-TR cells. As shown in Figure 5A, decreased cell viability caused by the treatment combination composed of paclitaxel and tephrosin was not restored by treatment with NAC (*p* > 0.05). The crystal violet analysis also showed that cell viability was effectively decreased by treatment with paclitaxel and tephrosin, regardless of NAC treatment (Figure 5B). It has been reported that changes in the expression of intracellular apoptosis factors such as the BCL-2 family regulate apoptosis and chemoresistance in cancer [41,42]. In Figure 5C, immunoblotting showed no significant difference in the expression of BCL-2, BCL-XL, MCL-1, BAX, or Survivin. Interestingly, the expression of the X-linked apoptosis protein (XIAP) was inhibited by the treatment with tephrosin alone and the combination treatment with paclitaxel and tephrosin. In addition, the expressions of cleaved caspase 3 and cleaved PARP were effectively increased using the combination treatment with paclitaxel and tephrosin. RT-PCR and RT-qPCR tests showed that there was no difference in the transcriptional expression of *BCL-2, BCL-XL, MCL-1, BAX, XIAP,* or *SURVIVIN* (Figure 5D,E). When the difference in the gene expression of *BCL-2, BCL-XL, MCL-1, BAX, XIAP,* and *SURVIVIN* was quantified, it was non-significant, as shown in Figure 5E (*p* > 0.05). These results indicated that tephrosin inhibits the expression of XIAP in paclitaxel-treated SKOV3-TR cells.

### 3.6. Tephrosin Also Restored Paclitaxel Resistance in Other Paclitaxel-Resistant Ovarian HeyA8-MDR Cells

To determine whether the mechanism of tephrosin-induced paclitaxel resensitization was a cell-specific phenomenon in SKOV3-TR cells, the cell viability assay was examined using HeyA8-MDR cells. Cells were treated with serially diluted concentrations of paclitaxel (0–500 nM) with dose-dependent concentrations of tephrosin (0, 5, and 10 µM) for 48 h. As shown in Figure 6A, the WST-1 analysis showed that the treatment combination composed of paclitaxel and tephrosin effectively induced cytotoxicity in HeyA8-MDR cells. The crystal violet analysis also identified the decreased cell viability of the paclitaxel and tephrosin combination treatment (Figure 6B). As shown in Figure 6C, tephrosin inhibited the FGFR1/FRS2 signaling pathway in HeyA8-MDR cells. These data suggested that tephrosin-induced paclitaxel resensitization was not a specific reaction for SKOV3-TR cells alone but that it also occurred in other paclitaxel-resistant ovarian cancer cell lines.

## 4. Discussion

Despite the fact that pharmaceutical research on anticancer drugs is active, paclitaxel is still used as the first-line treatment for ovarian patients [7]. It has been reported that the majority of recurrences of ovarian cancer occur due to chemoresistance to primary treatment [13]. Therefore, finding strategies for overcoming chemoresistance has emerged as one of the most important issues in ovarian cancer research. This study demonstrates that tephrosin effectively inhibits paclitaxel resistance by downregulating the FGFR1/FRS2 signaling pathway. These findings are expected to show the potential of tephrosin as a new target anticancer drug that can be applied in the treatment of paclitaxel-resistant ovarian cancer patients.

Tephrosin is one of the rotenoid families that is isolated from the *Derris, Lonchocarpus,* and *Tephrosia* species [80]. Rotenoid-based substances have long been used as phytochemicals due to their strong insecticidal activity [64]. It has been demonstrated that rotenone is a natural toxin that inhibits complex 1 of the mitochondrial electron transport chain, and chronic exposure to the strong toxicity of rotenone increases the risk of Parkinson’s disease [81]. On the other hand, a previous study showed that tephrosin has an anticancer effect and it contains less cell cytotoxicity than rotenone [68]. Figure 1A showed that cytotoxicity was not induced in up to 10 µM of treatment with tephrosin alone in SKOV3-TR cells. Previous reports have also suggested that tephrosin has anticancer activity against various cancer cells, including pancreatic cancer [68], lung cancer [69], and colon cancer cells [82]. However, the anticancer effects of tephrosin on ovarian cancer and related mechanisms of chemoresistance are still unclear. In this study, we demonstrated for the first time that tephrosin could suppress paclitaxel resistance in ovarian cancer cells.

The aim of our study is to reveal the effects of tephrosin on paclitaxel-resistant ovarian cancer cells. In lane 3 and 4 of Figure 2, the phosphorylation of AKT (S473), STAT3 (Y705), ERK (T202/Y204), and p38 MAPK (T180/182) was effectively inhibited by tephrosin. Studies have indicated that prosurvival signaling, such as AKT, STAT3, ERK, and p38 MAPK is one of the crucial mechanisms of paclitaxel in ovarian cancer [17]. Yang YI et al. have reported that tectorigenin increased the paclitaxel sensitivity of paclitaxel-resistant human ovarian cancer cells through the downregulation of the AKT and NF-κB signaling pathways [29]. MiR-181c was also identified to improve the paclitaxel sensitivity of ovarian carcinoma cells through the PI3K/AKT pathway [83]. Previous studies have shown that the STAT3 pathway mediates paclitaxel resistance [84,85]. Sheng H et al. have demonstrated that the inhibition of the STAT3 pathway restores paclitaxel resistance in ovarian cancer by downregulating G6PD expression [86]. Previous studies have demonstrated that the inhibition of the ERK signaling pathway induces paclitaxel sensitivity in paclitaxel-resistant ovarian cancers [22,87]. Also, it has been reported that the p38 MAPK signaling pathway is related to paclitaxel resistance in ovarian carcinoma, meaning that blocking this pathway can promote cellular apoptosis [25]. Fan LL et al. have suggested that the octreotide–paclitaxel conjugate reverses paclitaxel resistance by downregulating the p38 MAPK signaling pathway [36]. Our results suggest that tephrosin inhibits prosurvival signaling, and it may also affect paclitaxel resistance in SKOV3-TR cells.

Receptor tyrosine kinase (RTK) pathways have been known to regulate intracellular prosurvival signaling [73,74]. There is an accumulating amount of evidence implying that the aberration of the EGFR mediates oncogenesis and paclitaxel resistance in various cancers, including lung adenocarcinoma [88], cervical cancer [89], and ovarian cancer [90]. Tephrosin has been reported to have anticancer effects by inducing the degradation and internalization of the EGFR in human colon cancer cells [82], suggesting that tephrosin possibly restored paclitaxel sensitivity by inhibiting the EGFR signaling pathway. Our data indicated that the expression of the EGFR and its phosphorylation at the Y1045 residue showed no significant difference in the combination treatment with paclitaxel and tephrosin (Figure 3A). Interestingly, the phosphorylation of FGFR1 (Y653/654) was dramatically inhibited in the treatment with tephrosin. Consistent with this finding, the phosphorylation of FRS2 (Y196) led to a decrease in tephrosin-treated SKOV3-TR cells (Figure 3A). Our data suggested that tephrosin suppresses the FGFR1/FRS2 signaling pathway in paclitaxel-resistant ovarian cancer cells. Although the defined mechanism of action of RTK by tephrosin needs to be further elucidated, this does not rule out that there may be differential action depending on cell types.

Previous studies have shown that the FGFR signaling pathway also mediates the progression and chemoresistance of cancer [91,92,93]. Helsten T et al. have reported that aberrations in the FGFR signaling pathway contribute to the development of cancer and the degree of the amplification of the FGFR is overexpressed in ovarian cancer [54]. AZD4547, an inhibitor of the FGFR family, has been identified to exhibit an antitumor effect on ovarian cancer cells [94]. Moreover, our previous study indicated that cell viability is effectively reduced when BGJ398, a pan-FGFR inhibitor, is treated in a sphere culture of epithelial ovarian carcinoma cells, in which the prosurvival pathway is overexpressed [95]. In particular, several studies have demonstrated that a correlation exists between the FGFR signaling pathway and chemotherapy [62,96]. Szymczyk, J. et al. identified that fibroblast growth factor 1 (FGF1) triggers the activation of AKT activation FGFR-overexpressed cancer cells, consequently affording protection against the effects of paclitaxel [96]. Paul M et al. reported that the effect of chemotherapy was increased in the combination treatment with paclitaxel and derazantinib, an inhibitor of FGFR1-3, on human gastric tumors [62]. Collectively, we expected that the inhibition of FGFR signaling by tephrosin is a possible target to reduce paclitaxel resistance in ovarian cancer cells.

Additionally, we observed that tephrosin downregulates the expression of FGFR1 alone, but not FRS2 (Figure 3A). It has been known that the binding of ligands to FGFRs leads to the activation of receptor dimerization and the subsequent activation of the intracellular signaling pathway. FGFRs undergo internalization and lysosomal degradation to terminate activated signals [97,98,99]. The decreased expression of FGFR1 in lane 3 and 4 of Figure 3 is a phenomenon that occurred at the time, showing the possibility that tephrosin can act directly on FGFR1 (Figure 3A). Moreover, we observed that tephrosin had no effect on the expression of FGFR2 (Appendix A), implying that tephrosin effectively acts on FGFR1. We cannot rule out the possibility that tephrosin may affect other FGFR family members; the exact mechanism of action is being studied further.

The clinical success of paclitaxel has been limited by chemoresistance in cancers, mainly caused by the overexpression of the drug efflux transporters of the ATP binding cassette (ABC) family [9]. It has been revealed that the intracellular expression of P-gp confers paclitaxel resistance in ovarian cancer [9,100]. Accordingly, previous studies have shown that the inhibition of P-gp is being proposed as a therapeutic option for paclitaxel resistance [101,102]. We investigated whether the mechanism of tephrosin-induced paclitaxel resensitization occurs by regulating the expression and function of P-gp (Figure 4). Immunoblotting and the fluo-3/AM assay indicated that tephrosin had no effect on the expression or function of P-gp. Our data exclusively rule out the possibility that tephrosin may affect paclitaxel resistance through a mechanism of P-gp. Meanwhile, several studies have reported that cell cytotoxicity is induced by an increase in ROS as the AKT signaling pathway is inhibited in cancer cells [103,104]. Additionally, it has been found that tephrosin induces apoptosis in human pancreatic cancer cells by increasing the production of ROS [68]. These reports led us to investigate whether the cytotoxic effect of tephrosin in paclitaxel-treated SKOV3-TR cells was due to the induction of intracellular ROS. NAC did not restore the decreased cell viability in a SKOV3-TR combination treatment composed of paclitaxel and tephrosin, suggesting that the cell cytotoxicity induced by the cotreatment with paclitaxel and tephrosin in SKOV3-TR cells was not caused by ROS (Figure 5A,B).

It has been reported that paclitaxel-induced cell cytotoxicity is regulated by the pro-apoptotic and anti-apoptotic BCL-2 family proteins [105]. It has also been shown that these pro-apoptotic and anti-apoptotic BCL-2 families are regulated by survival signaling mechanisms and affect paclitaxel resistance in various cancer types [106,107,108]. However, the expression of BCL-2 families in tephrosin-treated SKOV3-TR cells was not observed. Interestingly, anti-apoptotic factor XIAP was significantly reduced by tephrosin (Figure 5C). Previous studies have revealed that the increased expression of XIAP and Survivin activates the metastasis and chemoresistance of cancer [109,110,111]. In particular, Lai et al. reported that apoptosis caused by inhibiting the FGFR1 signaling pathway is related to the downregulation of XIAP, BCL-2, and Survivin in pancreatic ductal adenocarcinoma [112]. In ovarian cancer, there is evidence demonstrating that the expression of XIAP not only regulates drug-induced apoptosis but also mediates chemoresistance [113,114,115,116]. Previous studies related to the effects of paclitaxel on apoptosis showed that apoptosis is increased by the combination of the Smac N7 peptide and paclitaxel due to the downregulation of XIAP and Survivin [117]. Our study suggested the possibility that the downregulation of XIAP is involved in apoptosis through the combination treatment with paclitaxel and tephrosin. The precise mechanism of tephrosin in the inhibition of XIAP expression and its role in suppressing paclitaxel resistance in SKOV3-TR cells need to be explored to further elucidate the underlying mechanism.

In conclusion, the present study suggests that tephrosin is a potent therapeutic agent for paclitaxel-resistant ovarian cancer. In addition, the mechanism for tephrosin is based on the FGFR1/FRS2 signaling pathway without P-gp function or intracellular ROS level changes. According to reports that aberrated FGFR signaling occurs at a relatively high rate in gynecological cancers, including breast and ovarian cancer [54], tephrosin may be used as a drug to suppress paclitaxel resistance in malignant gynecological cancers.

## 5. Conclusions

We demonstrated that a treatment combination composed of paclitaxel and tephrosin can inhibit paclitaxel resistance in paclitaxel-resistant ovarian SKOV3-TR cells and HeyA8-MDR cells. Although tephrosin alone did not decrease the cell viability of paclitaxel-resistant ovarian cells, the combination treatment composed of paclitaxel and tephrosin effectively induced cell cytotoxicity and apoptosis. Moreover, tephrosin suppressed the phosphorylation of AKT, STAT3, ERK, and p38 MAPK via the FGFR1/FRS2 signaling pathway. Interestingly, the mechanisms of tephrosin-induced paclitaxel resensitization were not related to the function of P-glycoprotein or the cellular level of reactive oxygen species. Our study suggested that tephrosin, a chemical derived from natural products, can effectively modulate paclitaxel resistance in ovarian cancer with a combination treatment including paclitaxel.

## Figures and Tables

**Figure 1 biomedicines-11-03155-f001:**
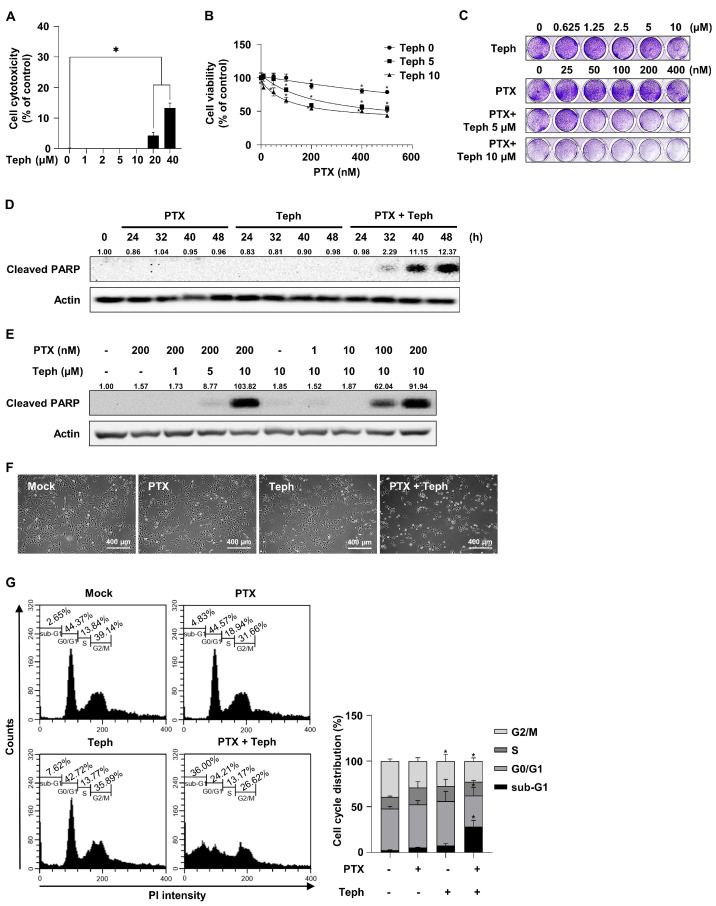
The cytotoxicity effect of combination treatment with paclitaxel and tephrosin in SKOV3-TR cells. (**A**) Cells (5.0 × 10^3^) were seeded in 96-well plates for 24 h. After treatment with tephrosin (0, 1, 2, 5, 10, 20, and 40 µM) for 48 h, the cell cytotoxicity was examined using the lactate dehydrogenase (LDH) assay. (**B**) SKOV3−TR cells (5.0 × 10^3^) were seeded in 96-well plates for 24 h. Cells were then treated with serially diluted paclitaxel (0–500 nM) with dose-varying combinations of tephrosin (0, 5, and 10 µM) for 48 h. Cell viability was measured using the water-soluble tetrazolium (WST-1) assay. (**C**) Cells (5.0 × 10^4^) were seeded in 24-well plates. After 24 h of incubation, cells were treated with serially diluted tephrosin (0–10 µM). Additionally, cells were treated with paclitaxel (0, 25, 50, 100, 200, and 400 nM) with dose-different combinations of tephrosin (0, 5, and 10 µM) for 48 h. The visualization of cell viability was determined using the crystal violet assay. (**D**) SKOV3-TR cells (6.0 × 10^5^) were seeded in 100 mm culture plates and then treated with dimethyl sulfoxide (DMSO) (mock), paclitaxel (200 nM), tephrosin (10 µM), and combination treatment with paclitaxel (200 nM) and tephrosin (10 µM) for 24 h, 32 h, 40 h, and 48 h, respectively. After the cells were harvested, the protein expression of cleaved PARP was analyzed using immunoblotting. Actin was used as a loading control. (**E**) Cells were treated with paclitaxel (200 nM) with dose-different combinations of tephrosin (0, 1, 5, and 10 µM) for 48 h. In addition, cells were treated with tephrosin (10 µM) with dose-different combinations of paclitaxel (0, 1, 10, 100 and 200 nM). Immunoblotting was used to determine the expression of cleaved PARP. (**F**,**G**) SKOV3-TR cells were treated with paclitaxel (200 nM), tephrosin (10 µM), and combination treatment with paclitaxel and tephrosin for 48 h. The morphological changes in the cells were observed using microscopy. Fluorescence-activated cell sorting analysis was performed to analyze the apoptotic fractions by co-treating with paclitaxel and tephrosin. Protein expression levels were quantified relative to the control, following normalization to the corresponding expression of actin using ImageJ software. All experiments were repeated three times. Significant differences were calculated using a one-way analysis of variance (ANOVA), and * *p* ≤ 0.05, as analyzed using the concentration, was considered significant. PTX, paclitaxel; Teph, tephrosin.

**Figure 2 biomedicines-11-03155-f002:**
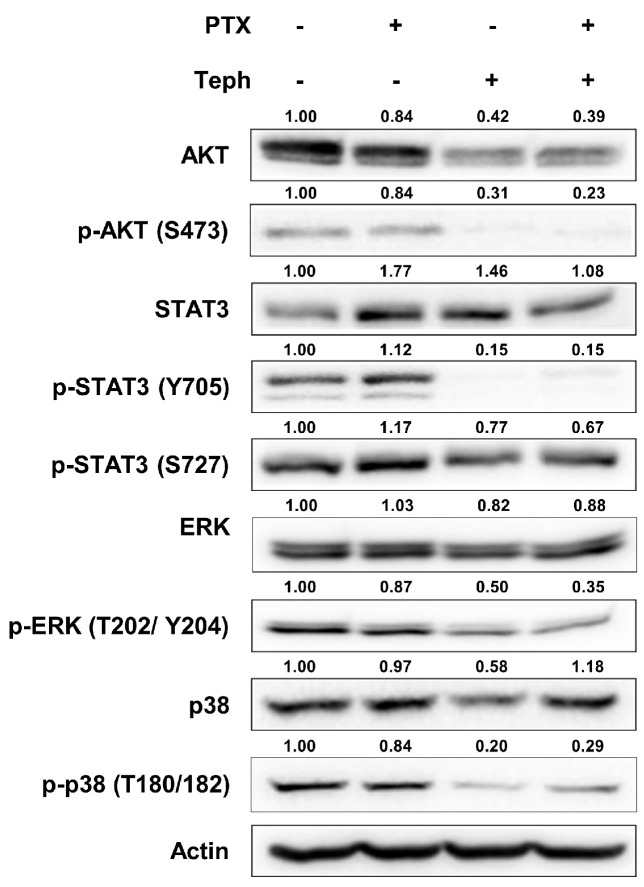
The effect of tephrosin on inhibition of phosphorylation of AKT, STAT3, ERK, and p38 MAPK in SKOV3-TR cells. SKOV3-TR cells (6.0 × 10^5^) were seeded in 100 mm culture plates for 24 h. Cells were then treated with DMSO (mock), paclitaxel (200 nM), tephrosin (10 µM), and combination treatment with paclitaxel and tephrosin for 48 h. After harvesting all the cells, AKT, STAT3, ERK, p38 MAPK, and their phosphorylation expression levels were analyzed using immunoblotting. Actin was used as a loading control. Protein expression levels were quantified relative to the control, following normalization to the corresponding expression of actin using ImageJ software. PTX, paclitaxel; Teph, tephrosin.

**Figure 3 biomedicines-11-03155-f003:**
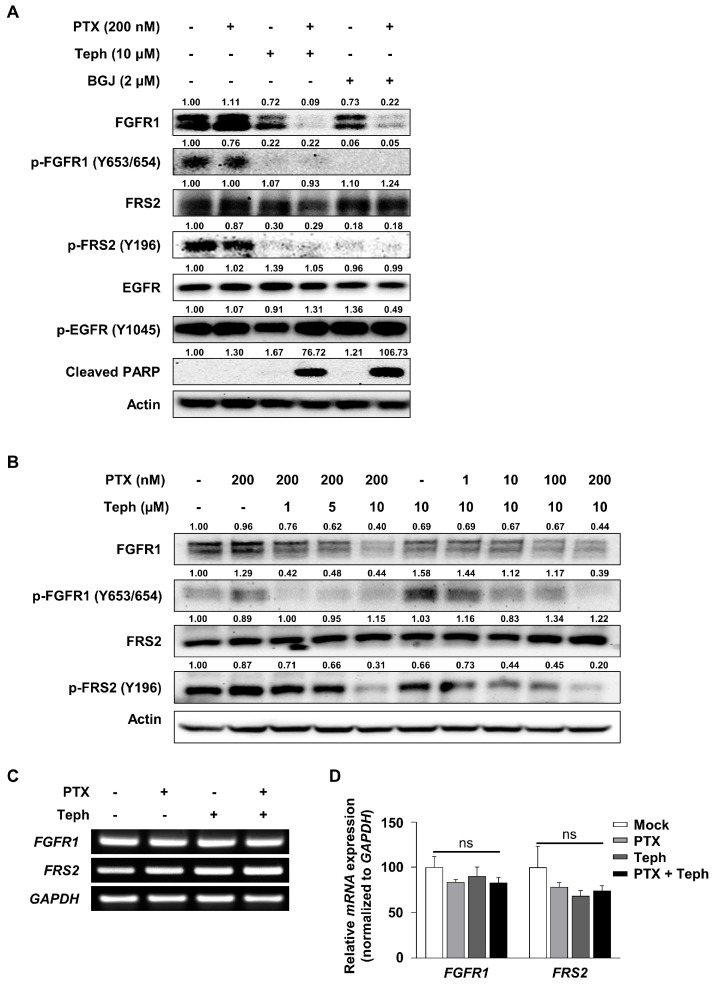
The effect of tephrosin on downregulation of the FGFR1/FRS2 signaling pathway in SKOV3-TR cells. (**A**) Cells were treated with DMSO (mock), paclitaxel (200 nM), tephrosin (10 µM), combination treatment with paclitaxel and tephrosin, BGJ398 (2 µM), and combination treatment with paclitaxel and BGJ398. After 48 h of treatment, immunoblotting was performed to determine the expressions of FGFR1, FRS2, EGFR, and their phosphorylation forms. The expression of cleaved PARP showed apoptotic cell death for each treatment condition. (**B**) SKOV3-TR cells were treated with paclitaxel (200 nM) with dose-different combinations of tephrosin (0, 1, 5, and 10 µM) for 48 h. In addition, cells were treated with tephrosin (10 µM) with dose-different combinations of paclitaxel (0, 1, 10, 100, and 200 nM). The expressions of FGFR1, FRS2, and their phosphorylation forms were analyzed using immunoblotting. Protein expression levels were quantified relative to the control, following normalization to the corresponding expression of actin using ImageJ software. (**C**) Cells were treated with paclitaxel and tephrosin, as shown in Figure 3A, and the mRNA expressions of *FGFR1* and *FRS2* were analyzed using RT-PCR tests. *GAPDH* was used as a loading control. (**D**) Relative quantifications of *FGFR1* and *FRS2* were analyzed using RT-qPCR tests, and the mRNA relative expression was normalized using *GAPDH*. Significant differences were calculated using a one-way analysis of variance (ANOVA). *p* > 0.05 indicated no significant (ns) difference. PTX, paclitaxel; Teph, tephrosin; BGJ, BGJ398.

**Figure 4 biomedicines-11-03155-f004:**
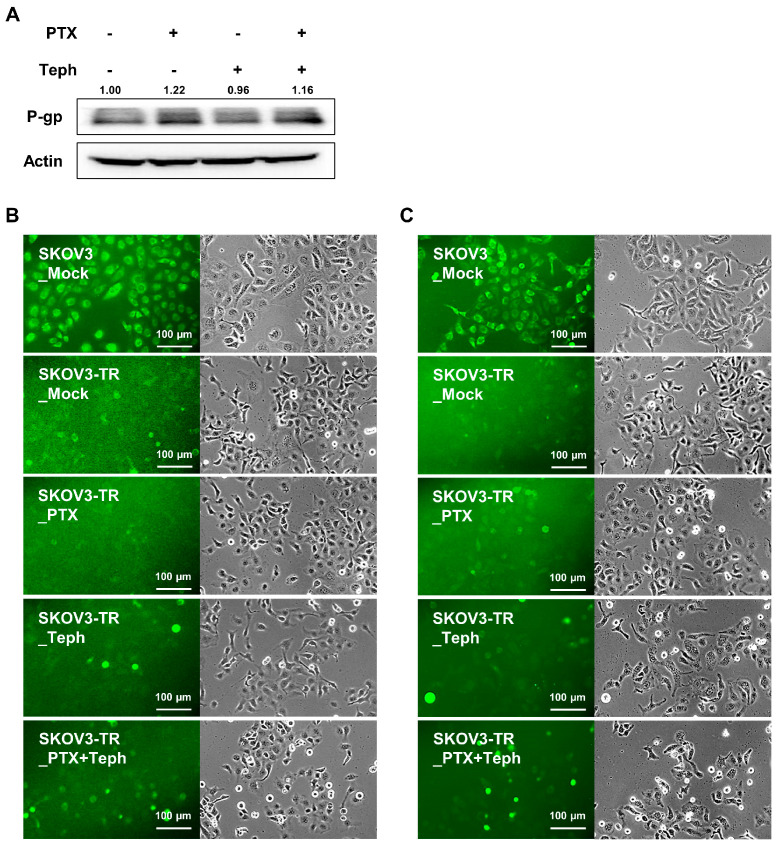
The non-significant effect of tephrosin on P-glycoprotein (P-gp) function in SKOV3-TR cells. (**A**) SKOV3-TR cells (6.0 × 10^5^) were seeded in 100 mm culture plates for 24 h and then treated with paclitaxel (200 nM), tephrosin (10 µM), and combination treatment with paclitaxel and tephrosin for 48 h. After all the cells were harvested, the expression of P-gp was determined using immunoblotting. Actin was used as a loading control. Protein expression levels were quantified relative to the control, following normalization to the corresponding expression of actin using ImageJ software. (**B**,**C**) SKOV3 cells were treated with DMSO (mock), and SKOV3-TR cells were treated according to the treatment condition of (**A**). After 24 h (**B**) and 48 h (**C**) treatment, cells were incubated with a 4 µM fluo-3 acetoxymethyl solution for 1 h. The amount of fluorescence produced by each experimental group was compared through a fluorescence microscope. PTX, paclitaxel; Teph, tephrosin; P-glycoprotein, P-gp.

**Figure 5 biomedicines-11-03155-f005:**
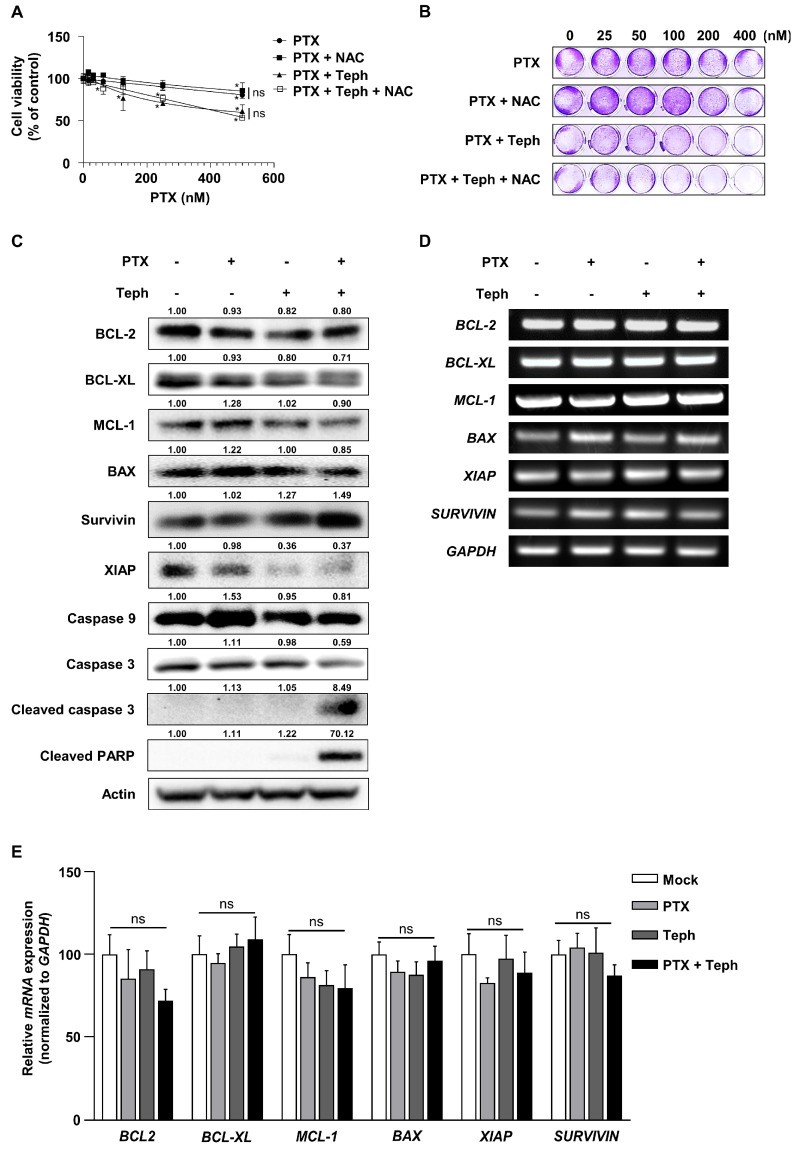
The effect of tephrosin on downregulation of XIAP expression and independent intracellular reactive oxygen species levels in SKOV3-TR cells. (**A**) SKOV3-TR cells (5.0 × 10^3^) were seeded in 96-well plates for 24 h. Cells were then treated with serially diluted paclitaxel (0–500 nM) with N-acetylcysteine (NAC) (5 mM), tephrosin (10 µM), and combination treatment with tephrosin and NAC for 48 h. Cell viability was measured using the WST-1 assay. (**B**) Cells (5.0 × 10^4^) were seeded in 24-well plates. After 24 h of incubation, cells were treated with paclitaxel (0, 25, 50, 100, 200, and 400 nM), with NAC (5 mM), tephrosin (10 µM), and combination treatment with tephrosin and NAC for 48 h. The visualization of cell viability was determined using the crystal violet assay. (**C**) Cells were treated with paclitaxel (200 nM), tephrosin (10 µM), and combination treatment with paclitaxel and tephrosin for 48 h. Immunoblotting was performed to analyze the expressions of BCL-2, BCL-XL, MCL-1, BAX, Survivin, XIAP, caspase 9, caspase 3, cleaved caspase 3, and cleaved PARP. Actin was used as a loading control. Protein expression levels were quantified relative to the control, following normalization to the corresponding expression of actin using ImageJ software. (**D**) Cells were treated with paclitaxel and tephrosin as shown in (**C**), and the mRNA expressions of *BCL-2, BCL-XL, MCL-1, BAX, XIAP,* and *SURVIVIN* were analyzed using RT-PCR tests. *GAPDH* was used as a loading control. (**E**) Relative quantifications of *BCL-2, BCL-XL, MCL-1, BAX, XIAP,* and *SURVIVIN* were analyzed using RT-qPCR tests, and the mRNA relative expression was normalized using *GAPDH*. All experiments were repeated three times. Significant differences were calculated using one-way ANOVA. * *p* ≤ 0.05, as analyzed using the concentration, was considered significant, and *p* > 0.05 indicated no significant (ns) difference. PTX, paclitaxel; Teph, tephrosin; NAC, N-acetylcysteine.

**Figure 6 biomedicines-11-03155-f006:**
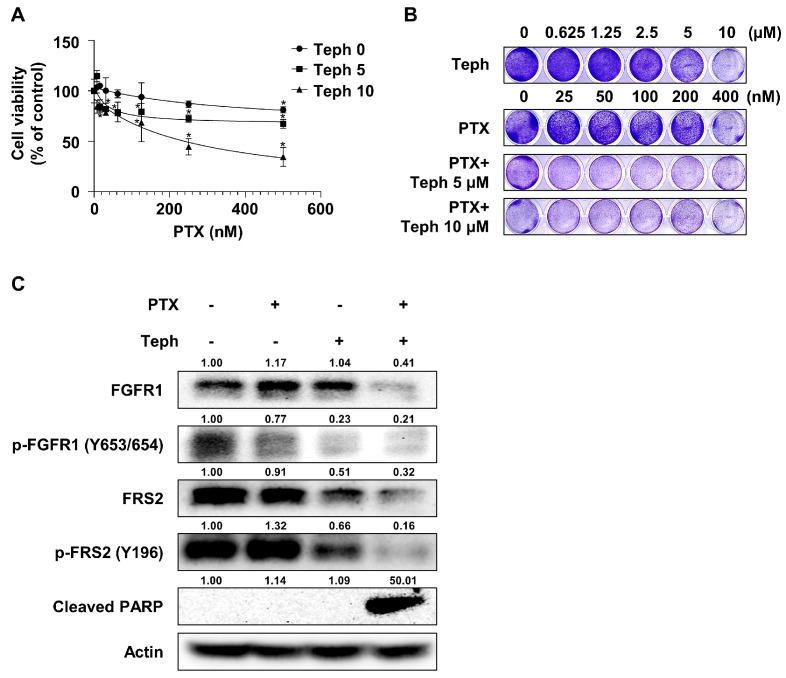
The cytotoxicity effect of combination treatment with paclitaxel and tephrosin in other paclitaxel-resistant ovarian cancer cell lines. (**A**) HeyA8-MDR cells were seeded in 96-well plates for 24 h. Cells were then treated with serially diluted paclitaxel (0–500 nM) with dose-varying combinations of tephrosin (0, 5, and 10 µM) for 48 h. Cell viability was measured through the WST-1 assay. (**B**) HeyA8-MDR cells were seeded in 24-well plates. After 24 h of incubation, cells were treated with paclitaxel (0, 25, 50, 100, 200, and 400 nM) with dose-different combinations of tephrosin (0, 5, and 10 µM) for 48 h. Additionally, cells were treated with serially diluted tephrosin (0–10 µM). The visualization of cell viability was determined using the crystal violet assay. (**C**) HeyA8-MDR cells were treated with DMSO (mock), paclitaxel (200 nM), tephrosin (10 µM), and combination treatment with paclitaxel and tephrosin for 48 h. Immunoblotting was performed to determine the expressions of FGFR1, FRS2, and their phosphorylation forms. Actin was used as a loading control. Protein expression levels were quantified relative to the control, following normalization to the corresponding expression of actin using ImageJ software. All experiments were repeated three times. Significant differences were calculated using one-way ANOVA, and * *p* ≤ 0.05, as analyzed using the concentration, was considered significant. PTX, paclitaxel; Teph, tephrosin.

## Data Availability

Data are contained within the article and Appendix A.

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
