# Peer review of "Tephrosin Suppresses the Chemoresistance of Paclitaxel-Resistant Ovarian Cancer via Inhibition of FGFR1 Signaling Pathway"

_biomedicines, 2023, doi:10.3390/biomedicines11123155_

Round 1

Reviewer 1 Report

Comments and Suggestions for Authors

Resistance to chemotherapy is a persistant problem especially to paclitaxel and it is important to find novel therapeutic solutions including natural products.

This is an excellent study showing that one such product, the natural rotenoid isoflavonoid,Tephrosin, supresses the chemoresistance to paclitaxel and elucidated the pathway and signaling steps through which this occurs: a downregulation of survival signaling via FGFR1 and the phosphorylation of AKT, STAT3, ERK, and p38 MAPK.

It is a excentally designed and complete study and very well written and presented. Since paclitaxel (and nab-paclitaxel) are utilized in tumers other tahn ovarian, these results have possibility to translated to those tumors and thus thai is a very important study.

I throughly support the publication of this manuscript

Author Response

Thank you sincerely for reviewing and evaluating our manuscript. We are very pleased to have the opportunity to improve our work.

Reviewer 2 Report

Comments and Suggestions for Authors

The manuscript “Tephrosin suppresses the chemoresistance of paclitaxel-resistant ovarian cancer via inhibition of FGFR1 signaling pathway” by Hee Su Kim and co-authors to find that tephrosin, a natural rotenoid isoflavonoid, can resensitize paclitaxel in paclitaxel-resistant ovarian cancer cells. Tephrosin inhibited the phosphorylation of AKT, STAT3, ERK, and p38 MAPK, which play important roles in survival signaling pathways simultaneously. Notably, tephrosin downregulated the phosphorylation of FGFR1 and its specific adapter protein, FRS2, but had no effect on the phosphorylation of EGFR. Immunoblotting and fluo-3 acetoxymethyl assay showed that tephrosin did not affect the expression or function of P-glycoprotein. Additionally, treatment with N-acetylcysteine did not re store cell cytotoxicity caused by combination treatment with paclitaxel and tephrosin, showing that tephrosin did not affect the reactive oxygen species scavenging pathway. Interestingly, tephrosin reduced the expression of the anti-apoptotic factor XIAP. This study demonstrates that tephrosin is a potent antitumor agent against paclitaxel-resistant ovarian cancer via inhibition of FGFR1 signaling pathway. However, some concerns that must be taken into account before the work can be reconsidered for publication.

Comment

1.      The data of western blot should be quantified. The mRNA data should be confirmed by real-time PCR.

2.      Can you please provide of the IC50 value of tephrosin in SKOV-3 PTX sensitivity cells? How about the resensitize paclitaxel in SKOV-3 PTX sensitivity cells?  

     3. Figures 1G, 1B and 5A: The P value should be provided.

Comments on the Quality of English Language

Extensive editing of English language required.

Author Response

저희 원고를 검토하고 평가해 주셔서 진심으로 감사드립니다. 우리는 업무를 개선할 수 있는 기회를 갖게 되어 매우 기쁩니다. 서론과 결과를 주의 깊게 검토한 후, 귀하의 권고에 따라 원고를 수정했습니다. 그러니 개정된 원고와 첨부파일을 보시고, 본 연구에 대해 의견을 보내주시면 감사하겠습니다.

Round 2

Reviewer 2 Report

Comments and Suggestions for Authors

The revised manuscript “Tephrosin suppresses the chemoresistance of paclitaxel-resistant ovarian cancer via inhibition of FGFR1 signaling pathwayhave adequately addressed my previous concerns and the paper is now acceptable for publication.